# Ex Vivo Osteoclastogenesis from Peripheral Blood Mononuclear Cells Is Unchanged in Adults with Phenylketonuria, Regardless of Dietary Compliance

**DOI:** 10.3390/ijms26125776

**Published:** 2025-06-16

**Authors:** Beatrice Hanusch, Anne Schlegtendal, Thomas Lücke, Kathrin Sinningen

**Affiliations:** 1Research Department of Child Nutrition, University Children’s Hospital of Ruhr-University Bochum, St. Josef-Hospital, Ruhr-University Bochum, 44791 Bochum, Germany; 2Department of Paediatric Pulmonology, University Children’s Hospital of Ruhr-University Bochum, St. Josef-Hospital, Ruhr-University Bochum, 44791 Bochum, Germany; 3Department of Neuropediatrics and Metabolism, University Children’s Hospital of Ruhr-University Bochum, St. Josef-Hospital, Ruhr-University Bochum, 44791 Bochum, Germany

**Keywords:** aromatic amino acids, phenylalanine, bone turnover, osteoclasts, dietary compliance, oxidative stress response, congenital metabolic disease

## Abstract

Pathogenic variants in the phenylalanine hydroxylase gene can result in phenylalanine (Phe) accumulation leading to phenylketonuria (PKU; OMIM #261600), a metabolic disease diagnosed in newborn screening. Early treatment with a Phe-restricted diet prevents severe mental retardation. Next to several other health complaints, patients with PKU present with low bone mineral density (BMD) more often than the general population. The etiology of the phenotype is not yet fully understood, and current research focuses on improving special medical foods and changes in osteoclasts (OC) and osteoblasts. Analysis of osteoclastic and oxidative stress control gene expression next to the simple number of OC developing from peripheral blood mononucleated cells (PBMCs) in association with dietary compliance and BMD was therefore part of our analysis. PBMCs were obtained from 17 adults with PKU and 17 age- and sex-matched controls on the same day. PBMCs were differentiated into osteoclasts (OC, Trap-positive multi-nucleated cells (≥3 nuclei)) for 14 days by adding human macrophage colony-stimulating factor (MCSF) and receptor activator of NF-κB Ligand (RANKL). Subsequently, quantitative real-time PCR was performed on OC function and oxidative stress control. Data on dietary compliance during the previous 12 months and 5 years and BMD were collected. PBMCs from adults with PKU and controls were differentiated into comparable numbers of OC (PKU: 53 [17–87] vs. controls: 39 [19–52], *p* = 0.381) without differences in mRNA expression of genes related to OC function and oxidative stress control. Dietary compliance in short-term and mid-term was not associated with OC number or mRNA expression, but *CTSK* negatively correlated with BMD T-Score in the hips of adults with PKU (Spearman r = −0.518, *p* = 0.040). Osteoclastogenesis was not changed in adult patients with PKU, nor were most mRNA expressions of OC marker genes or those of oxidative stress control. However, 44% of patients presented with BMD below −1 in their hips, and the OC of these tended to express higher *CTSK* (above −1: 0.2 [0.2–0.8] vs. below −1: 0.9 [0.6–3.4], *p* = 0.055). Thus, alternative regulatory mechanisms of OC activity may play a role in the development of low BMD in patients with PKU.

## 1. Introduction

In healthy humans, most of the ingested essential aromatic amino acid phenylalanine (Phe) is metabolized to tyrosine by phenylalanine hydroxylase (PAH) [1,2]. Pathogenic variants in the *PAH* gene can lead to reduced function of the enzyme, resulting in an accumulation of Phe and toxic byproducts of Phe in the blood, which can be detrimental to neural development [2,3]. In Germany, the severe form of PAH deficiency, phenylketonuria (PKU; OMIM #261600), was diagnosed in 6.2/100,000 newborns in 2021, whereas any hyperphenylalaninemia occurred in 15.1/100,000 newborns [3]. The main goal of the treatment of patients with PKU is to prevent severe mental retardation, for which the dietary intake of Phe must be reduced as early in life as possible [2]. Therefore, the intake of natural protein is reduced, and supplementation of Phe-free amino acids, vitamins, and minerals is needed [2]. During childhood, tight control of blood Phe concentration is necessary to ensure healthy development, which is loosened during adolescence, but lifelong treatment is recommended [2,4]. Due to significant impairment of quality of life in adolescents with PKU, therapeutic adherence tends to decline in these patients [2,5]. In 2018, French insurance claims of patients with PKU and matched controls were analyzed. While 40% of early-treated patients older than 16 years were prescribed Phe-free amino acid supplements (AAS), only 3% of late-diagnosed patients with PKU (all born before 1972) were treated with AAS [6,7].

Early treatment can prevent serious neurological consequences, but other health complaints occur. Next to chronic ischemic heart disease, asthma, diabetes, and osteoporosis, even in young patients, are diagnosed significantly more often in patients with PKU than in control patients according to German and French health insurance claims [6,8]. Lower bone mineral density (BMD) has been described in both early-diagnosed, well-treated and late-diagnosed, poorly treated children and adolescents with PKU, even without a family history of osteoporosis [9,10]. Adults with PKU also showed significantly reduced BMD Z-Scores in the lumbar spine (LS), femoral neck, proximal femur, and total body compared to the general population but without a higher proportion of fractures [11]. In a group of pediatric patients with PKU, variations in serum Phe levels correlated negatively with the spinal BMD [12].

As bone tissue is constantly being remodeled, reduced BMD can have two origins: lower bone mass accrual or elevated bone resorption. It has been discussed that patients with PKU have reduced bone mass accrual due to reduced intake of calcium and phosphorous from natural foods, but a deficiency of these minerals has not been found [13]. Additionally, it has been documented that patients with PKU are more often vitamin D sufficient than the general population [14,15,16]. While the BMD of patients with PKU stayed within reference ranges in some studies, other studies revealed a significantly lower BMD with changes in bone resorption and formation compared to the reference population [17,18,19]. To improve patients’ therapy, it is essential to understand the origin of bone loss. AAS are consumed in large quantities by patients with PKU, preferably for their whole life, and might be involved to some extent in the observed bone loss [20,21,22]. A current topic is the development of Phe-free AAS that might result in bone mass accrual and reduced loss of bone mass by reducing acid load and stimulating osteoblast activity [23,24].

Additionally, two studies have previously described elevated spontaneous osteoclastogenesis from peripheral blood mononucleated cells (PBMCs) of patients with PKU [25,26], while one study specifically found elevated bone resorption in adult patients with PKU without elevated bone formation [27]. While osteoclast (OC) precursors were elevated, higher levels of oxidative stress in osteoblasts were reported, leading to an imbalance of bone formation and resorption [25,26,28,29]. Analysis of osteoclastic and oxidative stress control gene expression next to the simple number of OC developing from PBMCs in association with dietary compliance and BMD was therefore part of our analysis.

## 2. Results

### 2.1. Study Population Characteristics and Dietary Adherence of Patients with PKU

Seventeen adult patients with PKU and seventeen age- and sex-matched controls participated in the study, with each patient–control pair being examined on the same day (PKU: 18.3–51.7 years, ♀ n = 10; Co: 18.3–54.9 years, ♀ n = 9; Table 1). Bone fractures were documented in the year prior to study inclusion in only one control, while none of the patients with PKU reported such incidents.

Twelve patients with PKU stated taking AAS, five of which took glycomacropeptides (GMP) and seven amino acid mixtures (AAM). One patient stated not taking any supplementation, and had not measured Phe concentrations during the last 5 years. One patient combined tetrahydrobiopterin (BH_4_) and AAM, while data on supplement usage was missing for four patients. Overall, mean blood Phe concentration during the 5 years prior to study participation was 611 µmol/L (± 345 µmol/L; min. 171 µmol/L, max. 1501 µmol/L, Table 1) in the 14 participants with data on blood Phe during the 5 years prior to study participation. During the 12 months prior to study participation, 12 patients with PKU had measured their Phe blood concentration (757 ± 375 µmol/L; min. 250 µmol/L, max. 1500 µmol/L, Table 1).

BMD in proximal femur (hip) was below a T-Score of −1 (T-Score_Hip_: −0.63 ± 1.07) in seven (44%) patients with PKU, the same was true for BMD in and at the lumbar spine (L1-L4; LS) (T-Score_LS_: −0.74 ± 1.14).

### 2.2. Number of Osteoclasts

After 14 days of differentiation, patients with PKU had a median of 53 (25–75th percentile: 17–87) OC per well, while controls had a median of 39 (25–75th percentile: 19–52) OC per well (Figure 1). The number of OC per well did not differ between patients with PKU and controls (*p* = 0.381, Figure 1). Additionally, no difference in OC count was observed between patients with good dietary compliance and those with poor dietary compliance during the last 5 years (good compliance 50.7 ± 31.3 cells vs. poor compliance 58.0 ± 49.2, *p* = 0.739, Table 2) and 12 months (good compliance 59.1 ± 27.1 cells vs. poor compliance 52.9 ± 49.1, *p* = 0.779, Table 2). For sensitivity analysis, patients with missing dietary compliance were excluded. No difference between dietary compliance groups in number of OC count was observed in this sensitivity analysis. Furthermore, no difference in the number of OC was found in patients with PKU with a T-Score in their hip or LS below −1 and above −1 (Table 3).

### 2.3. Gene Expression

Gene expression of OC function and oxidative stress control in OC did not differ between patients with PKU and controls (Figure 1).

Cells of patients with PKU who adhered to the dietary treatment in short-term and mid-term showed no differences in relative gene expression of OC function and oxidative stress control compared to patients who adhered poorly to their dietary treatment (Table 2). For sensitivity analysis, patients with missing dietary compliance were excluded. No differences between dietary compliance groups in relative gene expression of OC function and oxidative stress control were observed in this sensitivity analysis. Additional correlation analysis did not show any significant correlation of mean Phe blood concentration during the last 12 months or 5 years with any of the relative gene expressions.

Patients with PKU and BMD T-Score below −1 in their hips or LS and above −1 showed similar relative gene expressions of OC function and oxidative stress control in the OC, except for *CTSK*, which tended to show higher relative expression in the OC of patients with BMD T-Score below −1 in their hips (Table 3). In the additional correlation analysis, BMD T-Scores in hips correlated significantly with the relative gene expression of *CTSK* (Spearman r = −0.518, *p* = 0.040).

## 3. Discussion

With improvements in the diagnosis and treatment of PKU during the last decades, the most severe consequences of the disease are prevented, but several neurological and extra neurological symptoms still occur. One of these is a lower BMD already observed in young patients with PKU but with increasing frequency in the aging population of patients with PKU [17]. While, on the one hand, bone mass accrual might be impaired, multiple studies have also observed higher bone loss in patients with PKU, with two observing elevated osteoclastogenesis [17,25,26,27,30]. In 17 adults with PKU, we did not observe higher OC differentiation from PBMCs or elevated gene expression of OC function and oxidative stress control compared to 17 age- and sex-matched controls. Neither short- nor long-term dietary control nor BMD in the hips or LS was associated with the number of OC or gene expression, except for *CTSK*.

Previously, Porta et al. reported increased spontaneous osteoclastogenesis from PBMCs in 20 patients with PKU compared to controls [25]. While they did not use RANKL or MCSF for osteoclast differentiation, PBMCs from patients and controls developed a strikingly higher number of tartrate-resistant acid phosphatase (Trap)-positive multi-nucleated cells after 13 days of cell culture than we observed after 14 days. This might be due to age differences between the two participating groups or differences in counting cells [25]. On the one hand, counting of OC is not yet automated; hence, subjective evaluation of cells could influence the number counted. We therefore had a single scientist count all wells while blinded to the diagnosis of the participant. On the other hand, Porta et al. observed a significant correlation of blood Phe during the year prior to PBMCs’ isolation and the number of OC, which we were unable to observe in short-term (12 months) or mid-term (5 years) Phe blood concentration [25]. Because Porta et al. did not report the blood Phe ranges in their study, comparison to the patients analyzed here is not possible, but it might play a part in the predisposition of PBMCs to develop into OC [25]. Additionally, even though we were only able to include 17 patients with PKU and age- and sex-matched controls, Porta et al. only included 3 more patients and controls [25]. The small number of participants in this study is certainly a limitation, but it is possibly not the main reason for differences between previous results and the current ones. Roato et al. compared spontaneous as well as stimulated osteoclastogenesis in 40 patients with PKU and controls and found significantly higher OC counts in cell cultures from patients with PKU compared to controls in both setups [26]. While they added 30 ng/mL RANKL for differentiation to their cells, cells in this study were differentiated from PBMCs with 50 ng/mL RANKL [26]. Therefore, the numbers of OC are not directly comparable due to different RANKL concentrations; however, we could not observe a significant difference in osteoclastogenesis between controls and patients [26].

While Roato et al. observed a correlation of short-term blood Phe concentration and count of OC comparable to Porta et al., they also observed a significant correlation of OC count and age of patients [25,26]. In a study on bone resorption and formation in 46 patients with PKU between 4 and 38 years of age, higher bone resorption than in reference values determined by the authors was observed in 7- to 14-year-old patients as well as adults [27]. While a simple correlation of blood Phe concentration during the year prior to study participation and deoxypyridinoline in urine was noted, the association did not remain significant in multiple linear regression. However, a negative association of bone resorption and age remained [27]. As patients in the study of Roato et al. with blood Phe below 600 µmol/L were younger (aged 10.8 ± 7.8 years) than patients in our study, the observed correlation of blood Phe concentration and OC count might be caused by the age and maturity of the participants rather than their dietary control [26]. The same might be true for the higher number of OC counted by Porta et al., as patients and controls were 14.0 ± 7.1 years old and therefore mostly in childhood and puberty, while patients and controls in our analysis were exclusively adults (35.3 ± 11.7 years old). The elevated bone resorption observed in children and adults with PKU seems to be balanced by elevated bone formation during childhood, but this mechanism of compensation seems to be lost after maturation of the patients [27]. Consequently, differences in patients’ maturity in the studies by Roato et al. and Porta et al. and our analysis might influence not only the susceptibility of the neuronal system towards higher Phe concentrations but also that of the skeletal system [25,26].

We observed a trend towards higher mRNA expression of *CTSK* in OC from patients with PKU and BMD below −1 in the hips. *CTSK* encodes the Cathepsin K enzyme, which is expressed in OC for collagen degradation and its inhibition has become one target of osteoporosis and osteoarthritis treatment [31]. While osteoblasts and osteocytes can secrete Cathepsin K, solely OC function and not formation or survival or other bone cells seem to be affected by inhibition of the enzyme [31]. Furthermore, Cathepsin K is crucial for the development of actin rings by OC and thereby for enabling bone resorption [32]. Hence, *CTSK* knockout in mice leads to reduced bone resorption [32]. This is quite interesting, as we did not observe changes in the bone resorption markers of these patients with PKU [33], even though *CTSK* mRNA expression was slightly, but insignificantly, increased in OC from patients with reduced BMD. Millet et al. did find higher bone resorption marker concentration in their analysis of adults with PKU without correlation to BMD [27]. Longitudinal analysis of OC resorptive activity in patients with PKU might be helpful to find the optimal timeframe in which to start treatment in these patients to preserve BMD.

As a balance of bone formation and resorption is necessary for healthy bones, several factors leading to the uncoupling of OC from osteoblasts activity have been studied [34]. One of these factors might be reactive oxygen species (ROS). A mechanism of bone loss observed in postmenopausal osteoporosis is the reduction of osteogenesis while osteoclastogenesis is increased [35]. As Phe itself can oxidize glutathione in solution, and Phe and its metabolites were shown to increase ex vivo hippocampal thiobarbituric acid-reactive species in a time and concentration-dependent manner in rats, oxidative stress could be involved in osteopenia observed in some patients with PKU [36]. In fact, in both well-treated and poorly treated pediatric patients with PKU, higher plasma thiobarbituric acid-reactive species combined with lower total antioxidant reactivity and erythrocyte glutathione peroxidase activity than in healthy controls was described previously, without an association to Phe concentration [37]. Although we did not observe higher lipid peroxides or C-reactive protein concentration or lower antioxidative vitamins C and E in patients with PKU compared to healthy controls [33], tissue or cellular levels of ROS management might still be regulated differently [38]. BACH1 is a member of transcription factors involved in the production of ROS and indispensable for osteoclastogenesis [39,40,41]. As we did not observe differences in *BACH1* mRNA expression in PBMC-derived OC from patients with PKU and controls, this antioxidative system might not be affected in these OC under standard cell culture settings. Additional analysis of several other antioxidative systems would also be of interest in future studies on osteoclastogenesis in patients with PKU, as these systems are regulated by accelerated breakdown of proteins, phosphorylation or binding of inhibitors or activators rather than by changes in mRNA [42]. On another note, alternate mechanisms of bone resorption not affecting OC count and enzyme expression have been studied, like increased energy supply by increased ATP production [34]. Alternative mechanisms for bone loss due to the hyperactivity of OC might therefore also occur in patients with PKU, and further analysis of the mechanisms could help understand the phenotype observed in some of the patients. Interestingly, ex vivo analyzes of osteoclastogenesis from murine bone marrow macrophages showed increased bone resorption activity in cells with added aromatic acids [43]. Next to changes in osteoclast activity, osteoblasts might also be affected in PKU, showing reduced activity and heightened ROS accumulation in mouse and cell studies [28,29,38]. Future research on the effects of elevated Phe concentration on osteoclastogenesis and osteoblast function as well as the actual concentration of Phe in the bone and its impact on osteoblasts and OC would be of interest.

### Limitations

Even though OC analyzed in this study were differentiated from primary cells, the microenvironment of actual bones in patients and controls cannot be imitated ex vivo. Accumulation of amino acid metabolites like homocysteine in bone might influence the activity of osteoblasts and OC in this environment [44,45]. Additionally, communication of bone cells could not be evaluated in the monoculture of PBMCs-differentiated OC, but observation of this crosstalk would be of interest in understanding the process of reduced BMD in patients with PKU. Furthermore, as only relative expressions of mRNA and OC count were analyzed here, functional assays, such as resorption pit formation or TRAP activity analysis, might shed light on potential functional differences of OC of patients. Additionally, the cohort of 17 patients and controls was quite small, and subgroup analyses were therefore hard to conduct. Even though we collected information on GMP and AAM usage, the groups were too small to compare osteoclastogenesis. Previously, higher BMD and lower calcium excretion were described in patients using GMP compared to AAM [21,22,46,47]. Evaluating the influence of the AAS used on ex vivo osteoclastogenesis would therefore be of interest for future research. However, we matched patients and controls by age and sex and isolated PBMCs on the same day to minimize external differences between the groups, representing a methodological strength.

## 4. Materials and Methods

### 4.1. Subjects

Adult patients with PKU were recruited to participate during routine check-up at the outpatient clinic in the pediatric hospital in Bochum by word of mouth, advertisement in prints for interest groups, and posts on social media. All patients were asked to bring a friend, partner, or sibling as control. Missing controls (Co) were recruited by word of mouth. Age- and sex-matched controls were included in the study on the same day as the patients with PKU. Written informed consent was given by all participants. The study was approved by the Ethics Committee of the Ruhr-University Bochum (No. 20-7008, 28 September 2020) in accordance with the Declaration of Helsinki. Full data on bone-metabolism-related parameters and lifestyle factors are published elsewhere [33]. In the following analysis, only patients with PKU with PBMCs culture and same day matching control PBMCs culture were included. The amount of eligible data of the parameters varied due to missing data or small blood sample volumes in patients and controls.

### 4.2. Anthropometric Data, Lifestyle, and Nutrition

As previously described [33], dietary compliance of patients with PKU was determined based on average blood Phe concentrations during the last 5 years prior to study participation. Short-term dietary compliance was determined on the basis of blood Phe concentrations measured during the 12 months prior to study participation. As all participants were over 18 years old at the time of study participation, the target Phe levels during the 12 months and 5 years before recruitment were 120–600 µmol/L according to the European guidelines [2]. For evaluation, mean blood Phe values were categorized as “good compliance” if the mean blood Phe concentration was below 600 µmol/L or as “poor compliance” in cases of mean blood Phe concentration of 600 µmol/L or above or if no Phe measurement was performed during the period. Current use of AAS, including AAM and GMP, in patients with PKU was recorded via questionnaire.

BMD through Dual X-ray Absorptiometry (DXA) measured not earlier than one year prior to participation was acquired from the records of patients with PKU, expressed in T-Scores in the proximal femur (hip) and at the lumbar spine (L1-L4; LS). The DXA device used was Hologic Delphi C (Hologic Medicor GmbH, Berlin, Germany) standardized for gender, weight, and ethnicity by the provider. Fractures during the last 12 months prior to study participation were recorded [33].

### 4.3. Sampling and Biochemical Analyses

#### 4.3.1. Osteoclast Differentiation

Unfasted venous blood was drawn in ethylenediaminetetraacetic acid (EDTA) monovettes. From these whole blood samples, peripheral blood mononuclear cells (PBMCs) were isolated from buffy coats via the Ficoll Method (Pancoll density: 1.077 g/mL, Pan Biotech, Aidenbach, Germany). Then, 1 × 10^6^ cells/cm^2^ were plated and incubated in α-MEM Eagle (#P04-21500 Pan Biotech, Aidenbach, Germany) with 1% Penicillin (10,000 U/mL)/Streptomycin (10 mg/mL) (Pan Biotech, Aidenbach, Germany) and 10% fetal bovine serum (Gibco, Life Technologie GmbH, Darmstadt, Germany) and maintained at 37 °C and 5% CO_2_. For the differentiation of PBMCs into osteoclasts, plated cells were preincubated with 25 ng/mL recombinant human macrophage colony-stimulating factor (MCSF; PeproTech, Hamburg, Germany) for 3 days. Afterwards, 50 ng/mL of recombinant human receptor activator of NF-κB Ligand (RANKL; PeproTech, Hamburg, Germany) was added in addition to MCSF. The medium was changed every 2 to 3 days. After 14 days, cells in 96-well triplicates were stained for Trap using a commercially available kit (Acid Phosphatase Kit 387-A; Sigma-Aldrich, St. Louis, MO, USA) according to the manufacturer’s instructions. Large cells stained positive for Trap with ≥3 nuclei were defined as OC and counted by one scientist blinded to the diagnosis of participants. Representative pictures were taken using a BZ-X 810 microscope (Keyence, Neu-Isenburg, Germany).

#### 4.3.2. RNA Isolation, RT, and Real-Time PCR

Total RNA from cell culture was isolated on the same day as TRAP staining was performed by using the ReliaPrep™ RNA Cell Miniprep System according to the manufacturer’s instructions and reverse transcribed using GoScript™ Reverse Transcription Mix, Random Primers (both Promega, Madison, WI, USA). Quantitative real-time PCR was performed with GoTaq^®^ qPCR Master Mix and 25 ng cDNA using the manufacturer’s standard protocol (Promega, Madison, WI, USA). For internal standardization, the expression of beta-2-microglobulin (B2M) was analyzed, and the results were calculated through the ΔΔCT method and presented in x-fold increase relative to *B2M*. All applied primer sequences are given in Table 4.

### 4.4. Statistical Analyses

The power calculation was performed for the difference in bone mineral density between patients and controls, which was the primary hypothesis of the study [33]. The sample size of n = 20 for each group was calculated based on a previous publication by Schwahn et al. with a power of 0.90, α = 0.05, and Cohen’s d effect size = 1.07 (G*Power for Windows, version 3.1, Düsseldorf, Germany [33,48]).

The statistical software package IBM^®^ SPSS^®^ Statistics for Windows, version 29.0 (IBM Corp., Armonk, NY, USA), was used for the statistical analyses. Descriptive data were analyzed by the Chi-square test or, where applicable where applicable, Fisher’s exact test. The Shapiro–Wilk test was used to evaluate normal distribution in small groups. The QQ plots were used to test for normal distribution in groups larger than n = 10. As patients and controls were matched, comparison between each patient–control pair was performed using paired tests. Normally distributed data were analyzed using parametric tests (Student’s t-test). Non-normally distributed data were analyzed using non-parametric tests (Wilcoxon test, Mann–Whitney test). Correlation analysis was performed using Pearson’s correlation for normally distributed data and Spearman’s correlation for non-normally distributed data. Values of *p* < 0.05 were considered significant. Normally distributed data are presented as mean ± standard deviation (SD) and non-normally distributed data as medians [25–75th interquartile range].

## 5. Conclusions

In conclusion, we observed neither higher osteoclastogenesis, as previously described in younger patients with PKU, nor changes in the mRNA expression of genes related to osteoclast function and oxidative stress control in adults with PKU compared to controls. We did observe, however, a trend towards higher CTSK mRNA expression in patients with PKU and lower BMD. Due to the small number of patients included in this study and the non-significance of this finding, the result can only be considered exploratory in nature. Future studies on actual Phe concentration in bone and its effect on OC and osteoblasts could facilitate a better understanding of the origin of low BMD and may result in effective prevention and treatment of low BMD. Since some patients with PKU show lower BMD than the general population with no clear etiology, the prevention of osteoporosis might be necessary. Differences in food composition and susceptibility to higher levels of Phe might play a part in this phenotype, although it is still unclear whether changes in either osteoclasts or osteoblasts are involved in the development of low BMD in these patients.

## Figures and Tables

**Figure 1 ijms-26-05776-f001:**
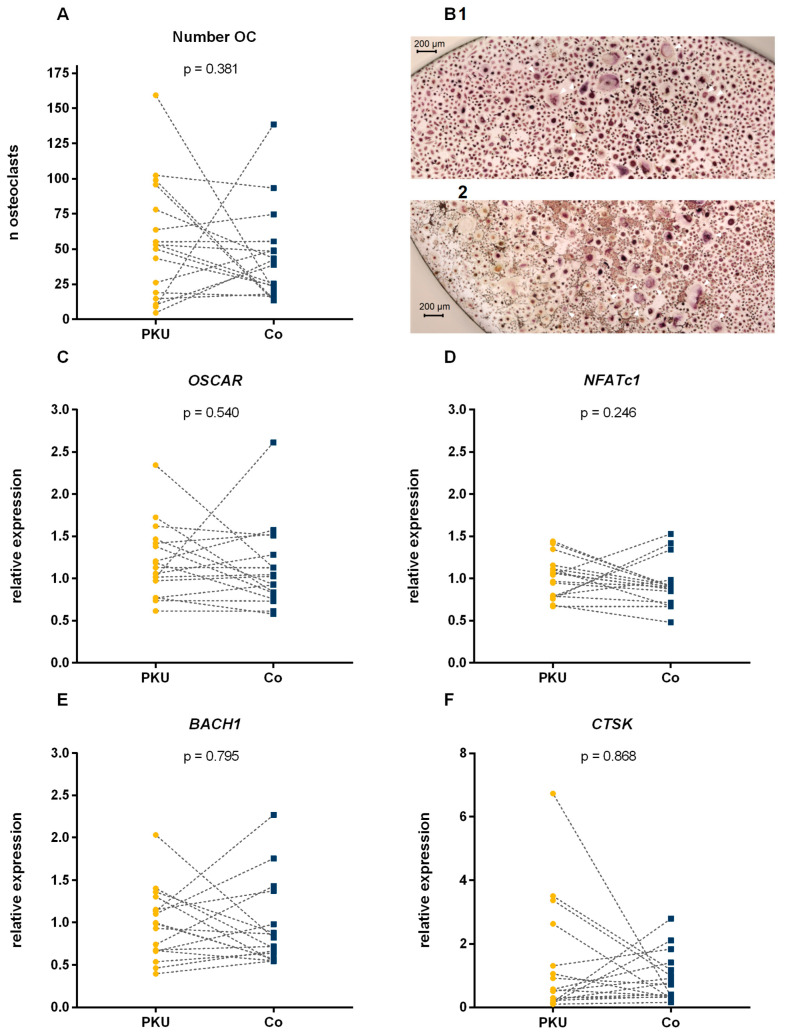
Cell count and relative expression of osteoclast specific and antioxidative genes in osteoclasts differentiated from PBMCs of patients with phenylketonuria (PKU) and controls (Co). Numbers of tartrate-resistant acid phosphatase (Trap)-positive multi-nucleated cells after 14 days of differentiation (**A**) and exemplary images (40× magnification (**B**)) of Trap-positive cells of a patient with PKU (1) and a control (2); white arrows mark representatives of osteoclasts (Trap-positive, ≥3 nuclei) at different stages of development. Relative mRNA expression of *Osteoclast-associated receptor* (*OSCAR*, (**C**)), *Nuclear factor of activated T cells 1* (*NFATc1*, (**D**), *BTB domain and CNC homolog 1* (*BACH1*, (**E**)) and *Cathepsin K* (*CTSK*, (**F**)). Dotted lines indicate patient–control pairs. Analyzed using paired Student’s t-test and paired Wilcoxon test.

**Table 1 ijms-26-05776-t001:** Characteristics of the participating patients with phenylketonuria (PKU) and controls (Co).

Parameter	PKU	Co	*p*
n	17	17	-
Age (years)	37.3 [23.4–44.5]	38.3 [23.8–47.2]	0.134
Female n (%)	10 (59%)	9 (53%)	0.500 ^a^
GMP usage n (%)	5 (41.7%)	-	-
Blood Phe 5 years (µmol/L)	610.6 ± 344.7	-	-
Blood Phe 5 years below 600 µmol/L n (%)	7 (41.7%)	-	-
Blood Phe 12 months (µmol/L)	756.7 ± 374.8	-	-
Blood Phe 12 months below 600 µmol/L n (%)	6 (35.3%)	-	-
T-Score in hip	−0.63 ± 1.07	-	-
T-Score in LS	−0.74 ± 1.14	-	-

^a^ chi square test; GMP, glycomacropeptides; LS, lumbar spine; Phe, phenalyalanine. Data are reported as median [25–75 percentile] (non-normal distribution) or as mean ± standard deviation (normal distribution).

**Table 2 ijms-26-05776-t002:** Comparison of count and gene expression of OC of patients with phenylketonuria with regard to their short-term and mid-term dietary control.

	Diet Compliance 12 Months		*p*	Diet Compliance 5 Years		*p*
	Good	Poor		Good	Poor	
Number of osteoclasts	59.1 ± 27.1	52.9 ± 49.1	0.779	50.7 ± 31.3	58.0 ± 49.2	0.739
*CTSK*	0.37 [0.13–1.64]	0.57 [0.30–3.37]	0.256	0.22 [0.14–2.63]	0.31 [0.74–1.86]	0.133
*OSCAR*	1.30 ± 0.56	1.11 ± 0.37	0.486	1.30 ± 0.51	1.14 ± 0.38	0.463
*NFATc1*	0.95 ± 0.18	1.04 ± 0.27	0.470	0.95 ± 0.19	1.05 ± 0.28	0.413
*BACH1*	0.67 [0.50–1.06]	1.10 [0.93–1.31]	0.122	0.93 ± 0.55	1.00 ± 0.32	0.734

Good diet compliance: mean blood Phe during the 5 years or 12 months prior to study participation below 600 µmol/L; poor diet adherence: mean blood Phe during the 5 years or 12 months prior to study participation ≥ 600 µmol/L or no measurement of Phe during the timeframe. *Cathepsin K* (*CTSK*), *Osteoclast-associated receptor* (*OSCAR*), *Nuclear factor of activated T cells 1* (*NFATc1*), *BTB domain and CNC homolog 1* (*BACH1*). Data are reported as median [25–75 percentile] (non-normal distribution) or as mean ± standard deviation (normal distribution).

**Table 3 ijms-26-05776-t003:** Comparison of count and gene expression of OC of patients with phenylketonuria with regard to bone mineral density in their hips and lumbar spine (LS).

	Hips		*p*	LS		*p*
	T-Score Above −1	T-Score Below −1		T-Score Above −1	T-Score Below −1	
Number of osteoclasts	51.0 ± 47.2	53.6 ± 35.8	0.903	50.7 ± 47.8	54.0 ± 34.7	0.882
*CTSK*	0.23 [0.15–0.79]	0.92 [0.56–3.37]	0.055	0.31 [0.20–2.04]	0.91 [0.30–2.63]	0.470
*OSCAR*	1.15 ± 0.34	1.24 ± 0.58	0.719	1.15 ± 0.34	1.24 ± 0.58	0.696
*NFATc1*	0.96 ± 0.23	1.08 ± 0.28	0.380	0.92 ± 0.23	1.13 ± 0.25	0.103
*BACH1*	0.74 [0.50–1.33]	1.00 [0.93–1.15]	0.536	0.92 ± 0.39	1.09 ± 0.47	0.457

*Cathepsin K* (*CTSK*), *Osteoclast-associated receptor* (*OSCAR*), *Nuclear factor of activated T cells 1* (*NFATc1*), *BTB domain and CNC homolog 1* (*BACH1*). Data are reported as median [25–75 percentile] (non-normal distribution) or as mean ± standard deviation (normal distribution).

**Table 4 ijms-26-05776-t004:** qPCR-primers used for analysis of osteoclast function and oxidative stress control in osteoclasts.

Protein	Gene	Sequence (5′ → 3′)		NCBI Reference Sequence
beta-2-microglobulin	*B2M*	F:	TAGCTGTGCTCGCGCTACTCTCTC	NM_004048.4
		R:	AATGTCGGATGGATGAAACCCAGACAC	
Cathepsin K	*CTSK*	F:	CCCGCAGTAATGACACCCTT	NC_000001.11
		R:	AAAGCCCAACAGGAACCACA	
Osteoclast-associated receptor	*OSCAR*	F:	CACTCCGTCTGTGGCCATTA	NM_206818.4
		R:	AGGACACATCCCGGAAGAGA	
Nuclear factor of activated T cells 1	*NFATc1*	F:	GTCCGTCTGTATGCGAGCAA	NM_172390.3
		R:	GGCTGCAACGGCGGAAGAAA	
BTB domain and CNC homolog 1	*BACH1*	F:	GTTTGTGGCTGGGGAGAGAAGG	NM_206866.3
		R:	ATGTTGTCGGGAAGTTCAGTGG	

## Data Availability

The data presented in this study are available upon request from the corresponding author.

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
