# Peer review of "Ex Vivo Osteoclastogenesis from Peripheral Blood Mononuclear Cells Is Unchanged in Adults with Phenylketonuria, Regardless of Dietary Compliance"

_ijms, 2025, doi:10.3390/ijms26125776_

Round 1

Reviewer 1 Report

Comments and Suggestions for Authors

See report attached

Comments on the Quality of English Language

See report attached

Reviewer 2 Report

Comments and Suggestions for Authors

The manuscript addresses an important clinical issue, bone health in people with PKU, focusing on the molecular aspect. The topic is relevant to IJMS, the study design is clear, and data are systematically presented. However, some areas need refinement to improve clarity and interpretation:

  • "It has been shown that Phe levels affect osteoblast and osteoclast activity in cell culture."  It would be beneficial to provide a brief example or mechanism of how Phe affects these cells if this is known.
  • "Differences between PKU patients and healthy controls were assessed by Student’s t-test."  Was a normality test performed before applying the t-test? If data are not normally distributed, a Mann-Whitney U test would be more appropriate.
  •  It would be beneficial to include Table summarizing the demographic and clinical characteristics of all included groups in the Results. This should include sex, age, BMD data if available, and average Phe levels for the PKU groups. This would facilitate review of the results because currently, age and Phe levels are only mentioned in the text
  • Besides the mentioned limitations, other potential limitations include the exclusively ex vivo setup, which might not reflect the complexity of the in vivo environment
  • Figure 1 should be reorganized to enhance its clarity 
